# Extended-spectrum beta-lactamase-producing Enterobacterales in human health: Experience from the tricycle project, Ghana

Noah Obeng-Nkrumah[1], Appiah Korang-Labi[2], Paul Kwao[1], Beverly Egyir[3], Benjamin D. Nuertey[4], George Hedidor[5], Gifty Boateng[6], Kwaku Asah-Opoku[7], Thomas Dankwah[8], Esther Okine[8], Japheth A. Opintan[2]*

1 Department of Medical Laboratory Sciences, School of Biomedical and Allied Health Sciences, College of Health Sciences, University of Ghana, Accra, Ghana, 2 Department of Medical Microbiology, University of Ghana Medical School, College of Health Sciences, University of Ghana, Accra, Ghana, 3 Bacteriology Department, Noguchi Memorial Institute for Medical Research, University of Ghana, Accra, Ghana, 4 Community Health Department, University of Ghana Medical School, College of Health Sciences, University of Ghana, Accra, Ghana, 5 World Health Organization- Ghana Office, Ghana, 6 National Public Health and Reference Laboratory, Korle Bu, Ghana, 7 Department of Obstetrics and Gynaecology, University of Ghana Medical School, College of Health Sciences, University of Ghana, Accra, Ghana, 8 Department of Microbiology, Korle Bu Teaching Hospital, Korle Bu, Ghana

* jaopintan@ug.edu.gh

**Data Availability Statement:** All relevant data are within the manuscript and its Supporting information files.

## Abstract

### Background

Vulnerable groups, such as pregnant women, are at increased risk of potentially life-threatening infections with extended-spectrum beta-lactamase-producing Enterobacterales (ESBL-E) for both mother and newborn. However, data regarding ESBL-E carriage and associated risk factors in Ghanaian pregnant women remain scarce.

### Objective

This study aimed to determine the prevalence of ESBL-E carriage and its associated risk factors among pregnant women attending the antenatal clinic at the Korle Bu Teaching Hospital.

### Methods

A systematic sample of 700 pregnant women with gestational age $\geq$ 34 weeks attending the antenatal clinic at Korle Bu Teaching Hospital was included in the study. After administering a structured questionnaire to assess potential risk factors associated with ESBL-E carriage, patients were given a sterile stool container to submit at least 1 g of stool specimen. Recovered isolates from faecal specimens were identified using MALDI-TOF-MS technology. These isolates were then subjected to susceptibility testing and ESBL identification. A random subset of 24 ESBL-producing *Escherichia coli* isolates was whole-genome sequenced on the MiSeq Illumina platform. Risk factors associated with ESBL-E carriage were determined using multivariable logistic regression analysis.

**Funding:** Fleming Fund support through World Health Organization country office, Ghana. There was no additional external funding received for this Study.

**Competing interests:** The authors have declared that no competing interests exist.

## Results

Among the 700 pregnant women, 42% (294) carried ESBL-E. The predominant ESBL-producing Enterobacterales were *Escherichia coli* (95%). Fifty percent (50%) of ESBL-E were multidrug resistant isolates (MDRs). Whole-genome sequencing of 24 ESBL-producing E. coli isolates revealed that blaCTX-M-15 (96%) was the most prevalent ESBL gene type. Notably, most isolates belonged to commensal phylogenetic groups (A, B1, and C; 88%). Having a primary level of education (aOR 1.45, 95% CI 1.05–1.96) and consuming legumes as the main source of protein (aOR 0.17, 0.40–0.83) were significantly associated with intestinal carriage of ESBL-E.

## Conclusion

This study identified a high prevalence of ESBL-E and MDR-ESBL-E carriage among pregnant women. Our findings underscore the urgent need for public health interventions to control the spread of AMR.

## Introduction

Treatment options for community-acquired Gram-negative infections are increasingly becoming limited as the frequency of community-acquired infections caused by Extended-Spectrum Beta-Lactamase producing-Enterobacterales (ESBL-E) increases [1]. Extended-spectrum beta-lactamases are enzymes that hydrolyse antibiotics containing a beta-lactam ring, including penicillins, broad-spectrum cephalosporins, and monobactams, except for carbapenems [2, 3]. ESBL-E often exhibit multi-drug resistance, affecting second-line antibiotic therapy for common community-acquired infections like urinary tract infections (UTIs) [4, 5]. The emergence of ESBL-E present a significant concern for vulnerable populations, particularly pregnant women who face a heightened risk of infections alongside their newborns [6, 7]. Some studies have reported reduced susceptibility to bacterial infections, including ESBL-E-associated urinary tract infections (UTIs) in pregnant women compared to their post-partum counterparts, potentially due to compromised immunity during pregnancy [6]. Colonization by ESBL-E in pregnant women is one of the key risk factors for infection with antibiotic-resistant bacteria (ARB) [8]. In addition, colonized pregnant women may also serve as a source of transmission to their newborns [9]. A critical knowledge gap exists in Ghana regarding the prevalence and associated risk factors for extended-spectrum beta-lactamase (ESBL) carriage among pregnant women. This lack of data is particularly concerning given the potential for vertical transmission of these resistant bacteria to newborns, possibly leading to adverse health outcomes. Closing this gap is critical, as improving maternal and child health outcomes is a key deliverable within the Sustainable Development Goals (SDGs) [10]. This study determined the prevalence of ESBL-E carriage and its associated risk factors among pregnant women attending the antenatal clinic at the Korle Bu Teaching Hospital.

## Methods

### Study design and setting

This was a cross-sectional surveillance study conducted between December 2021 and November 2022 at the antenatal clinic of the Obstetrics and Gynaecology Department of the Korle Bu

Teaching Hospital (KBTH). The KBTH is a 2000-bed tertiary referral healthcare facility in Ghana, with approximately 200 daily admissions. This Obstetrics and Gynaecology department houses 240 beds dedicated to obstetric care and 114 beds for gynaecological care. The department serves the antenatal clinics, labour wards, post-natal clinics, and the gynaecology unit. The antenatal clinic records an average daily attendance of 80 pregnant women, with about 70% (n = 56/80) of them having a gestational period $\geq$ 34 weeks, and the clinic operates five days a week. The study was well explained to the participants in their native language, allowing them to accept or decline the opportunity to be involved in this study.

## Study participants, sampling size and sampling procedure

Consenting pregnant women with gestational age $\geq$ 34 weeks attending the antenatal clinic at the Obstetrics and Gynaecology Department were included in the study. Pregnant women attending the antenatal clinic on account of obstetric and medical emergencies or exhibiting obvious signs of acute illness were excluded from the study. Using the guidelines from the ESBL-Ec tricycle project, 100 pregnant were recruited in 2021 and an additional 600 participants were recruited in 2022 [11, 12]. A systematic sampling technique was employed to recruit study participants. The sampling was done by selecting one sampling day each week. A starting day was selected from this day, and the next sampling day was shifted forward to the following day in the subsequent weeks. For example, if the starting point was Monday, the next sampling day was Tuesday the following week, Wednesday the next week, and so on. This sampling was iterated until the sample size was attained throughout the year. Selected participants provided informed written consent. Faecal specimens were collected, and a questionnaire was administered to evaluate risk factors for intestinal carriage of ESBL-E. A chronological overview of the study methods is provided in S1 Fig.

## Data collection tool

A standardized questionnaire was administered to collect participant demographics, lifestyle and hygienic practises and relevant clinical data towards the analysis of risk factors of ESBL-E carriage [1, 13, 14]. Data collected included socio-demographic data (e.g., age, gender, place of residence, marital status), patient lifestyle and hygienic practises (toilet facility at home, type of toilet facility, hand washing with soap after defecation, hand washing with soap before eating, frequency of bathing, use of treated drinking water), hospitalization history in the past year, antibiotic use within the past three and current months, gestational age and household size.

## Sample collection and laboratory investigation

Patients were given a sterile stool container and instructed to self-collect and submit at least 1 g of stool sample. Following sampling, the stool samples were transported to the laboratory within 2 hours after collection. Faecal samples were then inoculated on MacConkey agar (Sigma, UK) supplemented with 4 µg/mL cefotaxime (Sigma, UK) and incubated aerobically at 35–37 ˚C for 16–18 hours for the initial selection of 3$^{rd}$ generation cephalosporin-resistant isolates [15]. Discrete colonies of presumptive Enterobacterales isolates were tested for their reactions to routine biochemical reactions (indole test, urease test, citrate test, triple sugar iron test and motility test). Definitive identification of Enterobacterales was confirmed using the Bruker MALDI-TOF Biotyper (Bruker, USA). Phenotypic determination of ESBL production was performed using the combination disk diffusion method as recommended by the Clinical and Laboratory Standard Institute (CLSI) (2023) on Mueller Hinton agar [16]. This test was performed using ceftazidime (30 µg), cefotaxime (30 µg) and with and without clavulanate (10 µg), on Mueller Hinton Agar. *Klebsiella pneumoniae* ATCC 700603 was used as a positive

control for ESBL production. *Escherichia coli* ATCC 25922 was used as a negative control. All ESBL-positive and negative Enterobacterales from faecal specimens were tested for their antibiotic susceptibility using the Kirby-Bauer disk diffusion method according to guidelines by the CLSI (2023). The antibiotics were tested against the ESBL-E and Non-ESBL-E included: amoxiclav (30 μg), aztreonam (30 μg), meropenem (10 μg), tetracycline (30 μg), chloramphenicol (30 μg), ceftriaxone (30 μg), cefepime (30 μg), gentamicin (10 μg), amikacin (30 μg), ciprofloxacin (5 μg), meropenem (10 μg), and sulfamethoxazole trimethoprim (25 μg). *E. coli* ATCC 25922 was used as the control strain.

## Whole-genome sequencing

Twenty-four ESBL-*E. coli* were randomly selected by picking two isolates *per* month and whole genome-sequenced. Genomic DNA of the 24 ESBL-*E. coli* was obtained following the manufacturer's protocol for genomic DNA extraction (QIAamp DNA Mini and Blood Mini Handbook 3rd edition, 2010). Whole-genome sequencing was performed on a MiSeq Illumina sequencer (Illumina Inc., San Diego, CA, USA) using the NextEra DNA Flex Library Prep kit for library preparation (Illumina). Trimming of sequenced reads was performed using Trimmomatic v.3.0 [17]. *De-novo* assemblies of contigs were performed with Unicycler v0.5.0 [18]. Determination of acquired resistance genes and virulence genes was performed using ResFinder v4.2 [19], and VirulenceFinder v2.0 [20, 21]. Serotype and sequence type were determined using SerotypeFinder v2.0 [22] and pubMLST [23]. A maximum-likelihood phylogenetic tree was constructed using the CSIPhylogeny v1.4 [24] and tree annotation was performed using the Interactive Tree Of Life (ITOL) v6.8.1 [25]. Determination of phylogenetic groups of sequenced isolates was performed using ClermonTyping v23.06 [26, 27].

## Statistical analysis

Data were analysed using STATA IC-16. Comparisons between categorical data were conducted with $\chi^2$ or Fisher's exact tests. Point estimates of statistical significance were indicated with two-tailed $p < 0.05$. Univariate analyses were computed with an odds ratio (OR) with a 95% confidence interval (CI); variables with $p < 0.05$ were analysed in multivariate logistic regression models to determine independent associated predictor variables(s). The predictive accuracy of the models was evaluated by Hosmer and Lemeshow goodness-of-fit test with $p < 0.05$ suggesting that the model predicts accurately on average. The area under the Receiver Operating Characteristic Curve $> 0.7$ was used to analyse the discriminatory capability of ESBL faecal carriage versus their respective controls.

## Ethical approval

The study received ethical approval from the Institutional and Review Board of the Korle-Bu Teaching Hospital, before its commencement (***KBTH-STC/IRB/000168/2021***).

## Results

Overall, 761 pregnant women (110 in 2021 and 651 in 2022) who met the inclusion criteria were invited to participate in the study. Of these, 61 either did not consent, provide faecal samples, or could not participate in interviews and were consequently excluded. The remaining 700 women (100 in 2021 and 600 in 2022) consented, provided faecal samples, and participated in interviews for personnel data collection, resulting in a response rate of 91.8%. The mean age of pregnant women was 32 ± 5.50 years (range, 16–42 years) (S1 Table). Among the 700 participants, 40% (n = 280) had completed primary-level education, 29% (n = 206) had

**Table 1. Epidemiological metrics for indicators of antimicrobial resistance in faecal culture.**

| Indicator | | 2021 | 2022 | p-value |
|---|---|---|---|---|
| **Isolate-based metrics** | | | | |
| Prevalence of ESBL-*E. coli* among *E. coli* from faecal cultures | $\frac{number\ of\ ESBL-E.coli\ from\ faecal\ cultures}{Total\ number\ of\ 3rd\ gen.\ CrE.coli\ from\ cultures}*100$ | 57/68 = 83.8% | 222/281 = 79.0% | 0.891 |
| Prevalence of ESBL-Enterobacterales among Enterobacterales from faecal cultures | $\frac{number\ of\ ESBL-Enterobacterales\ from\ faecal\ cultures}{Total\ number\ of\ 3rd\ gen.\ CRE\ from\ cultures}*100$ | 57/68 = 83.8% | 237/296 = 80.1% | 0.709 |
| **Sample-based metrics** | | | | |
| Prevalence of patients with growth of ESBL-Ec | $\frac{number\ of\ sampled\ patients\ faecal\ culture\ with\ ESBL-E.coli}{Total\ number\ of\ patients\ with\ faecal\ cultures\ taken}*100$ | 57/100 = 57.0% | 222/600 = 37.0% | 3.782 |
| | $\frac{number\ of\ sampled\ patients\ faecal\ culture\ with\ ESBL-E.coli}{Total\ number\ of\ patients\ with\ ESBL\ faecal\ cultures\ for\ any\ probable\ isolates}*100$ | 57/57 = 100% | 222/237 = 93.7% | |
| | $\frac{number\ of\ sampled\ patients\ faecal\ culture\ with\ ESBL-E.coli}{Total\ number\ of\ patients\ with\ faecal\ cultures\ positive\ for\ 3rd\ gen.\ CRE.coli}*100$ | 57/68 = 83.8% | 222/281 = 79.0% | 0.891 |
| | $\frac{number\ of\ sampled\ patients\ faecal\ culture\ with\ ESBL-E.coli}{Total\ number\ of\ patients\ with\ faecal\ cultures\ positive\ for\ 3rd\ gen.\ CRE}*100$ | 57/68 = 83.3% | 222/296 = 75.0% | 1.551 |

* None of the faecal cultures was polymicrobial. P-value compares the independent proportions between 2021 and 2022 based on z scores from the chi-square test, 3rd gen. CRE.coli- Third-generation cephalosporins resistance Escherichia coli, 3rd gen. CRE- Third-generation cephalosporins resistance Enterobacterales, ESBL-Extended-spectrum beta-lactamase.

completed secondary-level education, 24% (n = 169) had completed tertiary-level education, and 7% (n = 45) had no formal education. Approximately 81% (n = 567) of the participants had a toilet facility at home, 78% (n = 551) had no history of hospitalization in the past year, while 92% (n = 647) had taken no antibiotics during the same period.

## Epidemiological metrics for indicators of antimicrobial resistance in faecal cultures

Overall, 364 third-generation cephalosporins-resistant Enterobacterales (3G-CrE) were recovered from the 700 faecal specimens cultured over two years. This included 349 (96%) *Escherichia coli*, 12 (3%) *Klebsiella pneumoniae* and 3 (1%) *Enterobacter cloacae*. Table 1 shows sample- and isolate-based metrics for ESBL-producing isolates. In 2021, the prevalence of third-generation cephalosporin Enterobacterales with growth of ESBL-producing Enterobacterales was 83.3% (n = 57/68) compared to 80.1% (n = 237/296) in 2022. Again in 2021, the prevalence of patient faecal specimens with growth of ESBL-producing *Escherichia coli* was 57.0% (n = 57/100) compared to 37.0% (n = 222/600) in 2022.

## Antimicrobial resistance patterns for third-generation cephalosporin resistance ESBL-positive and negative Enterobacterales

Generally, the proportion of isolates resistant to individual beta-lactam antibiotics was similar to the resistance level observed for non-beta-lactam antibiotics (Fig 1). For example, the mean percentage resistance of Enterobacterales to beta-lactams antibiotics was 54.8 ± 39.1% (range, 7–100%), compared to 52.2 ± 27.1% (range, 15–91%) for non-beta-lactam antibiotics (p = 0.167). When the susceptibility data was compared between ESBL-positive and ESBL negative Enterobacterales, the percentage resistance to each tested antibiotic did not differ significantly between the two groups (p > 0.05 for all pairwise comparisons). Approximately 46% (n = 169/364) of the Enterobacterales were MDR. The prevalence of MDR was relatively higher among ESBL-producing strains (50%, n = 147/294) compared to non-ESBL-producing strains

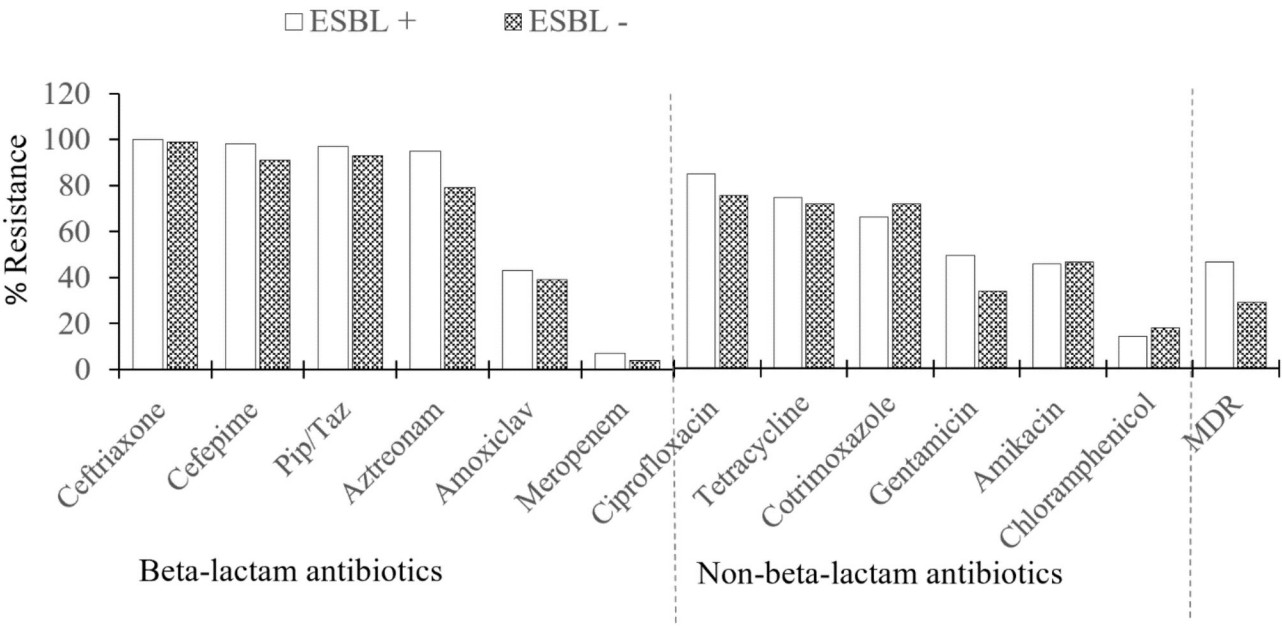

**Fig 1. Antimicrobial resistance patterns for ESBL-positive, ESBL negative, and MDR- isolates.**

(31%, n = 22/70). Overall, meropenem was the most effective antibiotic against both ESBL-positive and ESBL-negative MDR Enterobacterales. Among the non-beta-lactams, chloramphenicol recorded the lowest percentage of resistance to the isolates.

### Patient characteristics as risk factors for intestinal carriage of ESBL-E

In univariate analysis, adolescents, compared to adult pregnant women, were twice as likely to harbour ESBL-E in their intestines. Additionally, married women had slightly higher odds of ESBL-E carriage (OR 1.06) compared to single women (OR 0.94), but this difference was not statistically significant (Table 2). Pregnant women who reported consuming legumes as their primary protein source were significantly more likely to harbour ESBL-E in their intestines (p < 0.05). Additionally, pregnant women who did not wash their hands with soap after defecation had a higher risk of intestinal ESBL-E colonization (OR 1.08, 95% CI 1.00–1.17) compared to those who reported practising handwashing with soap after using the toilet. Multivariate logistic regression revealed that two variables were independently associated as risk factors for intestinal carriage with ESBL-E (Table 2). Primary level of education increased the risk of intestinal ESBL-E carriage by 45% (aOR, 1.45; 95% CI: 1.05–1.96). In contrast, the consumption of legumes as a main source of protein was protective against ESBL-carriage (aOR, 0.17; 95% CI: 04–0.83).

### Distribution of acquired antimicrobial resistance genes

Genomic analysis of the 24 *Escherichia coli* isolates revealed a heterogeneous gene distribution, leading to the identification of 21 acquired antimicrobial resistance genes using the ResFinder. Genes encoding resistance to beta-lactams (*CTX- M-15*, *CTX-M-27*, *TEM-1B*, *TEM-35*, *OXA-1*) and aminoglycosides *(aph (6)-1d*, *aph (3")-1b*, *aadA1*, *aac (3)-IIb*, *aadA2)* antibiotics were the most diverse, followed by folate inhibitors (*Sul1*, *Sul2*, *dfrA12*, *dfrA14*) and quinolones (*aac (6')-Ib-cr*, *qnrS1*, *qepA4*). The most common and predominant resistance genes harboured by

**Table 2.** Univariate and multivariate analysis of patient characteristics as possible risk factors for intestinal carriage of ESBL-E.

| Variable (n = 700) | Patient with ESBL-E Intestinal carriage | | Unadjusted odd Ratio (95% CI) | P-value | Adjusted odd Ratio (95% CI) | P-value |
|---|---|---|---|---|---|---|
| | No (406) | Yes (294) | | | | |
| **Social demographics** | | | | | | |
| Age | | | | | | |
| Adolescents | 5 | 2 | 1.82 (0.35–9.44) | 0.467 | | |
| Adults | 401 | 292 | 0.55 (0.11–2.85) | 0.467 | | |
| Educational level | | | | | | |
| None | 20 | 25 | 0.56 (0.30–1.02) | 0.060 | | |
| Primary | 177 | 103 | 1.43 (1.05–1.95) | 0.023 | 1.45(1.05–1.96) | 0.022 |
| Secondary | 114 | 92 | 0.86 (0.62–1.19) | 0.357 | | |
| Tertiary | 95 | 74 | 0.91 (0.64–1.29) | 0.589 | | |
| Married | 299 | 213 | 1.06 (0.76–1.49) | 0.725 | | |
| Employed | 98 | 81 | 0.84 (0.59–1.18) | 0.307 | | |
| Employment sector | | | | | | |
| Informal | 308 | 213 | 1.20 (0.85–1.68) | 0.307 | | |
| Private | 56 | 54 | 0.71 (0.47–1.07) | 0.102 | | |
| Public | 42 | 27 | 1.14 (0.69–1.90) | 0.611 | | |
| Tenancy | | | | | | |
| Single-tenant | 99 | 64 | 1.16 (0.81–1.66) | 0.419 | | |
| Multi-tenant | 307 | 230 | 0.86 (0.60–1.23) | 0.419 | | |
| Ownership(house) | | | | | | |
| Privately owned | 123 | 84 | 1.09 (0.78–1.51) | 0.622 | | |
| Public owned | 283 | 210 | 0.92 (0.66–1.28) | 0.622 | | |
| Travel overnight outside the home | 143 | 101 | 1.04 (0.76–1.42) | 0.812 | | |
| **Hygienic and lifestyle factors** | | | | | | |
| Toilet facility at home | 335 | 232 | 1.26 (0.83–1.84) | 0.231 | | |
| Type of toilet used | | | | | | |
| Flush | 268 | 187 | 1.11 (0.81–1.52) | 0.510 | | |
| Hole | 131 | 103 | 0.88 (0.64–1.21) | 0.444 | | |
| Mixed | 7 | 4 | 1.27 (0.37–4.39) | 0.703 | | |
| Toilet facility shared with others | 226 | 177 | 0.83 (0.61–1.12) | 0.231 | | |
| Hand washing before eating | 400 | 288 | 1.39 (0.44–4.35) | 0.573 | | |
| Hand washing after defecating | 403 | 292 | 0.92 (0.15–5.54) | 0.928 | | |
| Pipe water in the household | 306 | 222 | 0.99 (0.70–1.41) | 0.966 | | |
| Use of treated drinking water | 388 | 283 | 0.84 (0.39–1.80) | 0.651 | | |
| Use of boiled water | 10 | 3 | 2.45 (0.69–8.98) | 0.176 | | |
| Daily animal contact | 75 | 53 | 1.03 (0.70–1.52) | 0.880 | | |
| Primary source of protein | | | | | | |
| Fish | 288 | 207 | 1.02 (0.74–1.43) | 0.880 | | |
| Legumes | 2 | 8 | 0.18 (0.04–0.84) | 0.029 | 0.17(04–0.83) | 0.029 |
| Meat | 116 | 79 | 1.09 (0.78–1.52) | 0.620 | | |
| Frequency of bathing | | | | | | |
| Once | 1 | 3 | 0.36 (0.03–4.00) | 0.406 | | |
| Twice | 368 | 255 | 1.48 (0.92–2.38) | 0.105 | | |
| Thrice | 37 | 37 | 0.70 (0.43–1.13) | 0.142 | | |
| **Clinical factors** | | | | | | |
| Hospitalization in the past year | 89 | 60 | 1.09 (0.76–1.58) | 0.629 | | |

*(Continued)*

**Table 2.** (Continued)

| Variable (n = 700) | Patient with ESBL-E Intestinal carriage | | Unadjusted odd Ratio (95% CI) | P-value | Adjusted odd Ratio (95% CI) | P-value |
|---|---|---|---|---|---|---|
| | No (406) | Yes (294) | | | | |
| Stomach acid drug inhibitors | 1 | 2 | 0.36 (0.03–3.99) | 0.406 | | |
| Antibiotic use in the past year | 30 | 23 | 0.94 (0.53–1.65) | 0.830 | | |
| Hospitalization in the past three months | 41 | 40 | 0.71 (0.45–1.13) | 0.154 | | |
| Any surgical procedure in the past year | 16 | 8 | 1.47 (0.62–3.47) | 0.384 | | |

*CI- Confidence Interval

these isolates include *CTX-M-15* (n = 23/24), *qnrS1* (n = 17/24), *tetA* (n = 15/24) and *Sul2* (n = 14/24). Furthermore, all 24(100%) ESBL-positive isolates that harboured a particular ESBL gene; *CTX-M-15* (n = 23/24) and *CTX-M-27* (n = 1/24) demonstrated phenotypic resistance to ceftriaxone, cefepime and piperacillin-tazobactam. Also, 83% (n = 20/24) of the isolates that demonstrated phenotypic resistance to ciprofloxacin harboured a corresponding resistance gene(*qnrS1*), that encodes resistance to quinolones and 58% of the isolates that demonstrated phenotypic resistance to tetracycline and sulfamethoxazole-trimethoprim harboured a resistance gene that encodes resistance to tetracyclines (*tetA*) and folates inhibitors (*Sul2*) respectively (Fig 2).

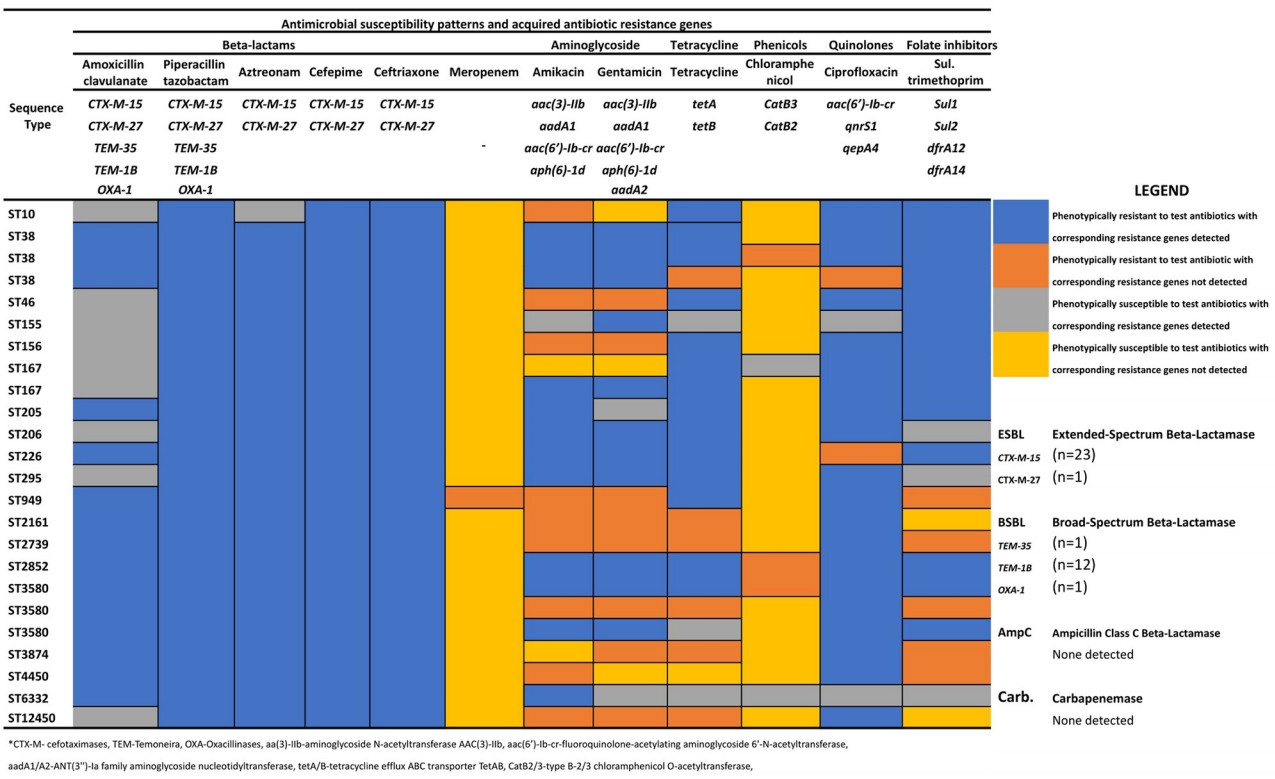

**Fig 2. Distribution of acquired antibiotic resistance genes and antibiotic susceptibility patterns.**

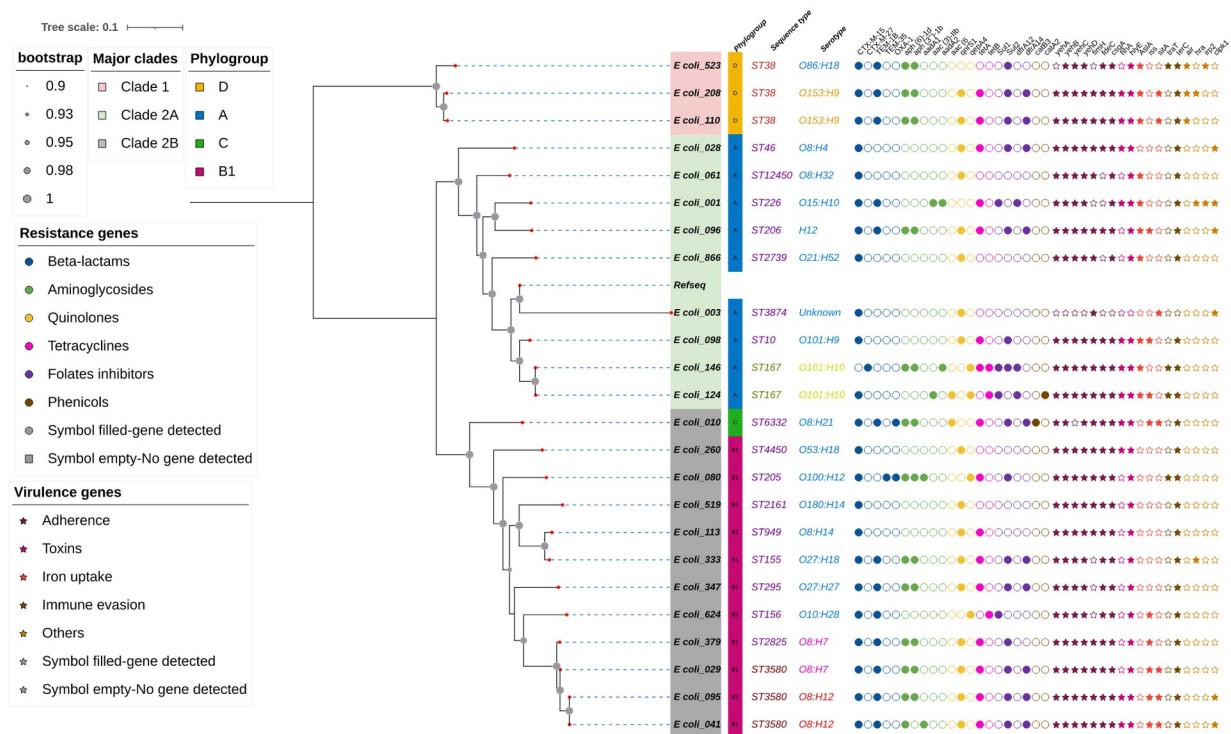

**Fig 3. Phylogenetic tree showing antimicrobial resistance and virulence genes according to sequence type and serotype.**

## Phylogenetic analysis

Phylogenetic tree was constructed using the CSIPhylogeny tool from the Centre for Genomic Epidemiology and Interactive Tree of Life (ITOL) tool to predict and annotate the evolutionary relatedness among the 24 ESBL-producing *Escherichia coli* and the frequently studied reference genome ASM584v2 (Escherichia coli str. K-12 substr. MG1655), based on core genome single nucleotide polymorphism (SNP) analysis. All 24 isolates were not closely related based on SNP difference of ≤ 10. Two (2) major clades were identified in this tree; Clade 1 and Clade 2, with Clade 2 giving rise to two major subclades 2A and 2B. Clade 1 comprises 3 taxa with 2 taxa having the same sequence type and serotype (ST38 & O153:H9). Clade 2A comprises 9 taxa with 2 sibling taxa of the same sequence type and serotype; ST167 & O101:H10. Also, Clade 2B comprises 12 taxa with 2 sibling taxa of the same sequence type and serotype; ST3580 & O8: H12 (Fig 3).

## Discussion

The results of this study showed that 42% of pregnant women were intestinally colonized with ESBL-E. This is similar to findings from Indonesia, Malawi, Chad and Madagascar which reported an estimated ESBL-E faecal carriage prevalence of 40%, 42% 45% and 56% respectively among pregnant women [12, 28–30]. This finding positions Ghana among the countries with the highest reported prevalence of ESBL-producing Enterobacterales (ESBL-E) colonization, aligning with the global trend of increasing ESBL-E carriage among community persons [28, 31]. Increasing prevalence of ESBL-E raises substantial concerns as it increases the risk of difficult-to-treat infections. Treatment of these infections may necessitate the use of last-resort

antibiotics, such as carbapenems, which in turn can contribute to the development and spread of carbapenemase-resistant strains [28, 29]. Relatively high prevalence of ESBL-E among pregnant women suggests that neonates born to these women stand a high risk of developing infections with ESBL-E.

Logistic regression analysis showed that a low level of education (primary education) was a significant risk factor for ESBL-E faecal carriage in most pregnant women, consistent with studies conducted by Djuikoue et al. (2016) and Watt et al. (200) [32, 33]. The observed association between low educational levels and increased ESBL-E carriage may be attributed to lack of awareness regarding hygiene and sanitation practices, both of which have been linked to the carriage of ESBL-E [34, 35]. This study revealed that 50% of ESBL-E were multi-drug resistant isolates, which is in line with a study conducted by Antony & colleagues which reported an overall 47% multi-drug resistance isolates among ESBL-E [36]. The occurrence of non-beta-lactam antibiotic resistance among ESBL-producing *Enterobacterales* could be explained by a multifaceted phenomenon involving plasmid-mediated resistance gene transfer, where plasmids carrying ESBL genes often harboured additional resistance determinants, cross-resistance mechanism, efflux pump systems, and selective pressures [8, 36–38]. This study further investigated the presence of acquired antibiotic resistance genes associated with ESBL production and other resistance determinants to non-beta-lactam antibiotics. The most common ESBL genes harboured by these isolates were $bla_{CTX-M-15}$ (96%, n = 23/24) and $bla_{CTX-M-27}$ (4%, n = 1/24). The most predominant non-beta-lactam resistance genes harboured by these isolates were *blaqnrS1* (70%, n = 17/24) conferring resistance to quinolones (ciprofloxacin), *blatetA* (63%, n = 15/24) conferring resistance to tetracyclines (tetracycline), *blaSul2* (58%, n = 14/24) conferring resistance to folate inhibitors and *blaaph(6)-1d*, *blaaph(3")-1b* (50%, n = 12/14) conferring resistance to aminoglycosides (gentamicin, amikacin). These results are similar to the findings of the study conducted by Milenkov et al. (2021) in Madagascar [29]. The presence of ESBL and non-beta-lactam resistance genes in these organisms poses a significant challenge in managing maternal and neonatal infections, potentially leading to limited therapeutic options and resulting in increased costs of patient management, morbidity and mortality rate [7, 39, 40].

Molecular analysis revealed that the 24 ESBL-producing *Escherichia coli* (ESBL-E) isolates belonged to a diverse set of phylogenetic groups. Multi-locus sequence typing identified 19 distinct sequence types (STs) clustered within 12 clonal complexes. Notably, 95% (n = 18/19) of the STs belonged to the commensal phylogenetic groups A, B1, and C, while the remaining 5% (n = 1/19) belonged to the pathogenic phylogenetic group D. These findings are similar to those from Madagascar and Chad, which reported a high prevalence of commensal phylogenetic groups A, B1, and C among ESBL-producing *E. coli* isolates from pregnant women with intestinal colonization [29, 30]. This finding could be explained by the fact that these isolates were isolated from one common source; the human intestinal tract where they are considered normal flora [41]. Consistent with the commensal nature of the isolated *E. coli* strains, our analysis revealed a limited number of virulence genes, primarily within the adhesin group, and a relatively low number of toxins detected. While most isolates belonged to the commensal phylogenetic group, which generally presents a lower risk of invasive disease, it is important to note that infections with commensal isolates are not uncommon [42, 43]. Also, considering their adept adaptation within the host's gut microbiota, ESBL-producing *Escherichia coli* (ESBL-*E. coli*) strains are likely to persist for extended periods, acting as significant reservoirs of resistance genes and posing a potential threat for human-to-human transmission [44]. To better understand ESBL-*E. coli* dissemination, we concentrated on isolates with a difference of less than 10 single nucleotide polymorphisms (n>10), which is the cut-off for closely related isolates in the literature according to Schürch et al. (2017) [45]. The core-genome-based SNP

analysis showed that there were no closely related isolates present in the sampling site. However, carriers of *E. coli*_041 (ST3580) and *E. coli*_095 (ST3580) with SNPs difference of 13 shared similar antenatal appointment dates, were previously admitted in the same hospital in the past three months and had the same number of antenatal visits (n = 8) suggesting a high likelihood of a common acquisition. Our study is not without limitations. First, the study was limited to only one tertiary hospital in Ghana, and the data is less likely to be a representation of what is happening in the entire country. Second, only a limited number of ESBL-positive Enterobacterales were whole-genome sequenced and analyzed, limiting the comprehensive representation of genetic diversity and evolutionary patterns present in the larger cohort of ESBL-E.

## Conclusion

This study identified a high prevalence of extended-spectrum beta-lactamase-producing Enterobacterales (ESBL-E) carriage among pregnant women. The study revealed a high rate of multi-drug resistance among the identified ESBL-E isolates, significantly limiting available antibiotic treatment options. These findings suggest a significant threat to both maternal and child health outcomes from ESBL-E, emphasizing the need for strict adherence to infection prevention and control strategies. This underscores the urgent need for public health interventions to control and combat the growing threat of antibiotic resistance in the country.

## Supporting information

**S1 Fig. Visualization of the data collection process and a concise summary of the study's results.** KBTH-Korle Bu Teaching Hospital; n-number; gest-gestation; obs&gynae dept-Obstetrics & Gynaecology department; gen-generation.
(TIF)

**S1 Table. Distribution of participants social demographics and clinical characteristics.**
SD-Standard deviation; n-number, min- minimum, max- maximum.
(PDF)

## Acknowledgments

We are grateful to the staff of the Antenatal clinic, Obstetrics and Gynaecology Department, Korle-Bu Teaching Hospital for their help in recruiting patients and collecting samples.

## Author Contributions

**Conceptualization:** Noah Obeng-Nkrumah, Appiah Korang-Labi, Beverly Egyir, Benjamin D. Nuertey, George Hedidor, Gifty Boateng, Kwaku Asah-Opoku, Japheth A. Opintan.

**Data curation:** Paul Kwao, Beverly Egyir.

**Formal analysis:** Noah Obeng-Nkrumah, Appiah Korang-Labi, Paul Kwao, Beverly Egyir, Gifty Boateng, Thomas Dankwah, Esther Okine, Japheth A. Opintan.

**Funding acquisition:** Noah Obeng-Nkrumah, Japheth A. Opintan.

**Investigation:** Paul Kwao, Japheth A. Opintan.

**Methodology:** Noah Obeng-Nkrumah, Appiah Korang-Labi, George Hedidor, Gifty Boateng, Kwaku Asah-Opoku, Thomas Dankwah, Esther Okine, Japheth A. Opintan.

**Project administration:** Japheth A. Opintan.

**Resources:** Japheth A. Opintan.

**Supervision:** Noah Obeng-Nkrumah, Appiah Korang-Labi, Beverly Egyir, Japheth A. Opintan.

**Validation:** Japheth A. Opintan.

**Writing – original draft:** Paul Kwao.

**Writing – review & editing:** Noah Obeng-Nkrumah, Appiah Korang-Labi, Beverly Egyir, Benjamin D. Nuertey, George Hedidor, Gifty Boateng, Kwaku Asah-Opoku, Thomas Dankwah, Esther Okine, Japheth A. Opintan.

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
