## [Decision Letter · Decision Letter 0]

2 Jul 2024

PONE-D-24-21155Extended-spectrum beta-lactamase producing Enterobacterals in human health: experience from the tricycle project, GhanaPLOS ONE

Dear Dr. Opintan,

Thank you for submitting your manuscript to PLOS ONE. After careful consideration, we feel that it has merit but does not fully meet PLOS ONE’s publication criteria as it currently stands. Therefore, we invite you to submit a revised version of the manuscript that addresses the points raised during the review process.

We look forward to receiving your revised manuscript.

Kind regards,

Mabel Kamweli Aworh, DVM, MPH, PhD. FCVSN

Academic Editor

PLOS ONE

Journal Requirements:

Fleming Fund support through World Health Organization country office, Ghana

"No"

5. Ethics statement appears in the Methods section of the manuscript AND at the end of the manuscript:

Your ethics statement should only appear in the Methods section of your manuscript. If your ethics statement is written in any section besides the Methods, please delete it from any other section. 

Additional Editor Comments:

In addition to addressing the reviewer's comments, please highlight the key limitations of the present study.

Reviewers' comments:

Reviewer's Responses to Questions

**Comments to the Author**

1. Is the manuscript technically sound, and do the data support the conclusions?

Reviewer #1: Yes

Reviewer #2: Yes

Reviewer #3: Yes

2. Has the statistical analysis been performed appropriately and rigorously? 

Reviewer #1: Yes

Reviewer #2: Yes

Reviewer #3: Yes

3. Have the authors made all data underlying the findings in their manuscript fully available?

Reviewer #1: Yes

Reviewer #2: Yes

Reviewer #3: Yes

4. Is the manuscript presented in an intelligible fashion and written in standard English?

Reviewer #1: Yes

Reviewer #2: Yes

Reviewer #3: Yes

5. Review Comments to the Author

**Reviewer #1:** Extended-spectrum beta-lactamase producing Enterobacterals in human health: experience from the tricycle project, Ghana

Accept after minor corrections on the methodologies used.

This was a job well-done!

Comments

Method

Lines 1-3. Information obtained from Korle Bu teaching hospital only is not reflective of the total population of Ghana, hence there might be a selection bias.

Data Collection

Lines 1-5, Does the standardized question obtain all the information required? Are there pieces of information that should be obtained that is not included in the structured questionnaire. Also, for patients that are uneducated, how does the team ensure that all the required data is obtained.

Additional Comments

1. Page 5: Method should be Methods since a number of methods were used.

2. For all methods used, authors used cite their previous authors, especially that these methods did not originate from them. All segments of the materials and methods except for Whole genome sequencing lack any reference.

3. Page 9: Result should be Results.

4. Page 9: This statement lacks a full stop, authors should add full stop:

"About 81% (n = 567) of the participants had a toilet facility at home, 78% (n = 551)

had no history of hospitalization in the past year, while 92% (n = 647) had not taken any antibiotics

during the same period"

5. Page 18: Conclusion, authors should identify any limitations to the execution of the research as this is necessary and obvious going by the methods used.

**Reviewer #2:** The manuscript is clear and well written. It addresses an important gap in knowledge regarding the

prevalence and associated risk factors for extended-spectrum beta-lactamase (ESBL) carriage

among pregnant women.

I have only one comment under the methodology section.

Methods

Study participants, sampling size and sampling procedure:

The authors mentioned......questionnaire was administered to evaluate risk factors for intestinal carriage of ESBL-E (S1_Fig). However, S1_Fig is a flow chart and not a questionnaire. Could you rectify this?

**Reviewer #3: **A very good job has been put into this research both in writing and technicalities hence few corrections I would like to highlight.

I would do the line count from each heading

1. Introduction: lines 3 -5 there is a disjoint between the sentences hence what you are trying to state is not clear; kindly rephrase or insert correct punctuation

2. I would like to know what informed your sample size and how you came about it

3.Sample collection and laboratory Investigation: Meropenem 20microgram was written in Line 17 and same in line 20. Kindly address

Thank you

6. PLOS authors have the option to publish the peer review history of their article (what does this mean?). If published, this will include your full peer review and any attached files.

Reviewer #1: **Yes: **Taiwo Akindahunsi

Reviewer #2: No

Reviewer #3: **Yes: **FOLASHADE ONATOLA TOYE

---

## [Author Response · Author response to Decision Letter 0]

4 Aug 2024

A separate sheet has been upload, addressing the reviewers comments

---

## [Decision Letter · Decision Letter 1]

20 Aug 2024

Extended-spectrum beta-lactamase producing Enterobacterals in human health: experience from the tricycle project, Ghana

PONE-D-24-21155R1

Dear Dr. Opintan,

We’re pleased to inform you that your manuscript has been judged scientifically suitable for publication and will be formally accepted for publication once it meets all outstanding technical requirements.

Kind regards,

Mabel Kamweli Aworh, DVM, MPH, PhD. FCVSN

Academic Editor

PLOS ONE

Additional Editor Comments (optional):

Please address the concerns of Reviewer 3 below:

On page 11 and page 12 esbl and esbl-e respectively , which are you intending to use? All capital letters or all small letters?

Reviewers' comments:

Reviewer's Responses to Questions

**Comments to the Author**

1. If the authors have adequately addressed your comments raised in a previous round of review and you feel that this manuscript is now acceptable for publication, you may indicate that here to bypass the “Comments to the Author” section, enter your conflict of interest statement in the “Confidential to Editor” section, and submit your "Accept" recommendation.

Reviewer #1: All comments have been addressed

Reviewer #2: All comments have been addressed

Reviewer #3: All comments have been addressed

2. Is the manuscript technically sound, and do the data support the conclusions?

Reviewer #1: Yes

Reviewer #2: (No Response)

Reviewer #3: Yes

3. Has the statistical analysis been performed appropriately and rigorously? 

Reviewer #1: Yes

Reviewer #2: (No Response)

Reviewer #3: Yes

4. Have the authors made all data underlying the findings in their manuscript fully available?

Reviewer #1: Yes

Reviewer #2: (No Response)

Reviewer #3: Yes

5. Is the manuscript presented in an intelligible fashion and written in standard English?

Reviewer #1: Yes

Reviewer #2: (No Response)

Reviewer #3: Yes

6. Review Comments to the Author

Reviewer #1: Reviewer's comment is also attached.

Review assignment for PONE-D-24-21155R1

PONE-D-24-21155R1

Extended-spectrum beta-lactamase producing Enterobacterals in human health: experience from the tricycle project, Ghana

Dear …….

Please find below my comments on the authors’ responses to the earlier reviewer’s comments:

1st Reviewers’ Comments Authors Response 2nd Reviewers’ Comments

Reviewer #1

Method Lines 1-3. Information obtained from Korle Bu Teaching Hospital only is not reflective of the total population of Ghana, hence there might be a selection bias We are grateful for the insightful comments. Indeed, we agree with the reviewer that Information obtained from Korle Bu Teaching Hospital only is not reflective of the total population of Ghana. This has been duly acknowledged as a limitation, to read, ‘Our study is not without limitations. First, the study was limited to only one tertiary hospital in Ghana, and the data is less likely to be a representation of what is happening in the entire country’ Satisfactory to the extent that the conclusion is not predicated on the population of Ghana.

Data Collection Lines 1-5, Does the standardized question obtain all the information required? Are there pieces of information that should be obtained that are not included in the structured questionnaire? Also, for uneducated patients, how does the team ensure that all the required data is obtained? All relevant patient characteristics/information determined to have association/possible association with intestinal carriage of ESBL-E among pregnant women from literature reviews and research with similar methodologies were obtained using the structured questionnaire. Notwithstanding, other additional pieces of information could have been added. Questionnaire was interpreted to un-educated study participants in their local/native languages by a trained personnel Satisfactory

Page 5: Method should be Methods since several methods were used. “Method” has been revised to “Methods” We are grateful to the reviewer. Satisfactory

For all methods used, authors cite their previous authors, especially since these methods did not originate from them. All segments of the materials and methods except for Whole genome sequencing lack any reference Comment duly addressed. Statements in the methodology section has been duly referenced with in-text citations “1, 11, 12, 13, 14, 15, 16” Satisfactory

Page 9: Result should be Results “Result” has been revised to “Results” Satisfactory

Page 9: This statement lacks a full stop, authors should add a full stop: "About 81% (n = 567) of the participants had a toilet facility at home, 78% (n = 551) had no history of hospitalization in the past year, while 92% (n = 647) had not taken any antibiotics during the same period" Comment duly addressed. The statement has been revised to include a full-stop Satisfactory

Page 18: Conclusion, the authors should identify any limitations to the execution of the research as this is necessary and obvious going by the methods used. Comment duly addressed. Some limitations were observed during the study and have been stated duly in the discussion. Satisfactory

Reviewer #2

Methods Study participants, sampling size and sampling procedure: The authors mentioned......questionnaire was administered to evaluate risk factors for intestinal carriage of ESBL-E (S1_Fig). However, S1_Fig is a flow chart and not a questionnaire. Could you rectify this? Comment duly addressed. The sentence has been revised to read, ‘A flow chart of how pregnant women were selected and processed is indicated in supplementary Figure 1 (S1 Fig).’ Satisfactory

Reviewer #3

Introduction: In lines 3 -5 there is a disjoint between the sentences hence what you are trying to state is not clear; kindly rephrase or insert the correct punctuation For clarity and logical flow, we have rephrase the statement to read, ‘Extendedspectrum beta-lactamases are enzymes that hydrolyse antibiotics with the betalactam ring, including penicillins, broad-spectrum cephalosporins, and monobactams except for carbapenems [2,3]. ESBL-E often exhibit multi-drug resistance, affecting secondline antibiotic therapy for common community-acquired infections like urinary tract infections (UTIs) [4,5]. Satisfactory

I would like to know what informed your sample size and how you came about it Given that there is no data on the prevalence of faecal carriage of ESBL-Ec in Korle-Bu Teaching Hospital or elsewhere in Ghana, we used a default value of 50% (0.5). The approach used here is consistent with what is provided in the literature. (Charan J, Biswas T. How to Calculate Sample Size for Different Study Designs in Medical Research? Indian journal of psychological medicine. 2013;35(2):121–6.) Satisfactory

Sample collection and laboratory Investigation: Meropenem 20 microgram was written in Line 17 and the same in Line 20. Kindly address The Meropenem disk concentration has been revised from 20 µg to 10 µg. We are most grateful to the reviewers. Satisfactory

Recommendation

The revised manuscript is hereby recommended for publication in view of the fact that the earlier comments have been addressed satisfactorily.

Acknowledgement

The opportunity to serve the Journal title is this capacity is deeply appreciated.

Dr. Taiwo Akindahunsi

Reviewer

Reviewer #2: (No Response)

Reviewer #3: I appreciate the effort taken to rewrite and make corrections where necessary to your write up.

On page 11 and page 12 esbl and esbl-e respectively , which are you intending to use? All capital letters or all small letters?

7. PLOS authors have the option to publish the peer review history of their article (what does this mean?). If published, this will include your full peer review and any attached files.

Reviewer #1: **Yes: **Taiwo Akindahunsi

Reviewer #2: No

Reviewer #3: **Yes: **FOLASHADE ONATOLA TOYE

---

## [Editor Report · Acceptance letter]

11 Sep 2024

PONE-D-24-21155R1 

PLOS ONE

Dear Dr. Opintan, 

I'm pleased to inform you that your manuscript has been deemed suitable for publication in PLOS ONE. Congratulations! Your manuscript is now being handed over to our production team.

Kind regards, 

on behalf of

Dr. Mabel Kamweli Aworh 

Academic Editor

PLOS ONE